# Calibrating dimension reduction hyperparameters in the presence of noise

**Justin Lin** [1] *, **Julia Fukuyama** [2]

**1** Department of Mathematics, Indiana University, Bloomington, Indiana, United States of America,
**2** Department of Statistics, Indiana University, Bloomington, Indiana, United States of America

* linjus@iu.edu

**Data Availability Statement:** All data and code are freely available at https://github.com/JustinMLin/ DR-Framework.

**Funding:** The author(s) received no specific funding for this work.

## Abstract

The goal of dimension reduction tools is to construct a low-dimensional representation of high-dimensional data. These tools are employed for a variety of reasons such as noise reduction, visualization, and to lower computational costs. However, there is a fundamental issue that is discussed in other modeling problems that is often overlooked in dimension reduction—overfitting. In the context of other modeling problems, techniques such as feature-selection, cross-validation, and regularization are employed to combat overfitting, but rarely are such precautions taken when applying dimension reduction. Prior applications of the two most popular non-linear dimension reduction methods, t-SNE and UMAP, fail to acknowledge data as a combination of signal and noise when assessing performance. These methods are typically calibrated to capture the entirety of the data, not just the signal. In this paper, we demonstrate the importance of acknowledging noise when calibrating hyperparameters and present a framework that enables users to do so. We use this framework to explore the role hyperparameter calibration plays in overfitting the data when applying t-SNE and UMAP. More specifically, we show previously recommended values for perplexity and n_neighbors are too small and overfit the noise. We also provide a workflow others may use to calibrate hyperparameters in the presence of noise.

## Author summary

In our infinitely complex world, perfect data rid of noise is an unattainable ambition. Hence, our goal is to coerce meaningful information, or the signal, from data inevitably riddled with unwanted, random variation. Advances in technology have allowed us to collect and process biological data of increasing size and complexity, so it is now more important than ever to acknowledge noise in our analyses to ensure random structures are not confused for significant patterns. Many algorithms and ideas have been suggested, some more cognizant of noise than others, but it is still unclear how noise should be handled in various situations. Our experiments, however, indicate typical calibrations of popular analysis methods are inadequately handling noisy, complex biological data. In response, we show and explain how alternate calibrations perform better in the presence of noise and lead to results more faithful to the data. By providing evidence of mishandled

**Competing interests:** The authors have declared that no competing interests exist.

noise and presenting solutions, we hope to further the discussion on handling noise in biological data.

## Introduction

In recent years, non-linear dimension reduction techniques have been growing in popularity due to their usefulness when analyzing high-dimensional data. Biologists use these techniques for a variety of visualization and analytic purposes, including exposing cell subtypes [1], checking for batch effects [2], and visualizing the trajectories of differentiating cells [3]. The most popular non-linear dimension reduction methods are t-distributed Stochastic Neighbor Embedding (t-SNE, [4]) and Uniform Manifold Approximation and Projection (UMAP, [5]). Both methods have been applied to various types of data within biology ([1, 6, 7]).

Since the introduction of t-SNE and UMAP, hyperparameter calibration has proven to be a difficult task. The most crucial hyperparameters, t-SNE's perplexity and UMAP's n_neighbors, control how large a neighborhood to consider around each point when determining its location in low dimension. Calibration is so troublesome, that perplexity-free versions of t-SNE have been proposed [8]. It is also an extremely important task, since both methods are known to produce unfaithful results when mishandled [9]. For t-SNE, the original authors suggested perplexities between 5 and 50 [4], while recent works have suggested perplexities as large as one percent of the sample size [10]. [11] studied the inverse relationship between perplexity and Kullback-Leibler divergence to design an automatic calibration process that "generally agrees with experts' consensus." For UMAP, the original authors make no recommendation for optimal values of n_neighbors, but their implementation defaults to n_neighbors = 15 [5]. Manual tuning of perplexity and n_neighbors requires a deep understanding of the t-SNE and UMAP algorithms, as well as a general knowledge of the data's structure.

The primary purpose of dimension reduction is to simplify data in a way that eliminates superfluous or nonessential information, i.e. noise. Each dimension reduction method does this slightly differently, but most require hyperparameter calibration. For example, the classical linear method, PCA, requires tuning of the number of principal components. A more contemporary method in biology, PHATE (Potential of Heat-diffusion for Affinity-based Trajectory Embedding) [3], requires tuning of a hyperparameter named diffusion time scale $t$. PHATE represents the structure of the data by computing local similarities then walking through the data using a Markovian random-walk diffusion process. $t$ determines the number of steps taken in a random walk and "provides a tradeoff between encoding local and global information in the embedding" [3]. Perplexity and n_neighbors serve the same purpose in their respective algorithms. Hence, we believe t-SNE and UMAP are capable of handling noise, but naïve calibrations that disregard noise often result in overfitting.

To assess dimension reduction performance in the presence of noise, we must acknowledge noise during the evaluation process. When the data's structure is available, we can visualize the results and choose the representation that best captures the hypothesized structure. In supervised problems, for example, we look for low-dimensional representations that cluster according to the class labels. For unsupervised problems, however, the structure is often unknown, so we cannot visually assess each representation. In these cases, we must resort to quantitative measures of performance to understand how well the low-dimensional representation reproduces the high-dimensional data. While this strategy is heavily discussed in the machine learning literature, many prior works disregard the possibility of overfitting when quantitatively measuring performance.

In this paper, we present a framework for studying dimension reduction methods in the presence of noise (Section 3). We then use this framework to calibrate t-SNE and UMAP hyperparameters in both simulated and practical examples to illustrate how the disregard of noise leads to miscalibration (Section 4). We also discuss how other researchers may use this framework in their own work (Section 5) and present a case study that walks the reader through the application of the framework to a modern data set (Section 6).

## Background

### t-SNE

t-distributed Stochastic Neighbor Embedding (t-SNE, [4]) is a nonlinear dimension reduction method primarily used for visualizing high-dimensional data. The t-SNE algorithm captures the topological structure of high-dimensional data by calculating directional similarities via a Gaussian kernel. The similarity of point $x_j$ to point $x_i$ is defined by

$$p_{j|i} = \frac{\exp(-||x_i - x_j||^2/2\sigma_i^2)}{\sum_{k \neq i} \exp(-||x_i - x_k||^2/2\sigma_i^2)}.$$

Thus for each point $x_i$, we have a probability distribution $P_i$ that quantifies the similarity of $x_i$ to every other point. The scale of the Gaussian kernel $\sigma_i$ is chosen so that the perplexity of the probability distribution $P_i$, in the information theory sense, is equal to a pre-specified value also named perplexity,

$$\text{perplexity} = 2^{-\sum_{j \neq i} p_{j|i} \log_2 p_{j|i}}.$$

Intuitively, perplexity controls how large a neighborhood to consider around each point when approximating the topological structure of the data. As such, it implicitly balances attention to local and global aspects of the data with high values of perplexity placing more emphasis on global aspects. For the sake of computational convenience, t-SNE assumes the directional similarities are symmetric by defining

$$p_{ij} = \frac{p_{i|j} + p_{j|i}}{2n}.$$

The $p_{ij}$ define a probability distribution $P$ on the set of pairs $(i, j)$ that represents the topological structure of the data.

The goal is to then find an arrangement of low-dimensional points $y_1, \ldots, y_n$ whose similarities $q_{ij}$ best match the $p_{ij}$ in terms of Kullback-Leibler divergence,

$$D_{KL}(P||Q) = \sum_{i,j} p_{ij} \log \frac{p_{ij}}{q_{ij}}.$$

The low-dimensional similarities $q_{ij}$ are defined using the t distribution with one degree of freedom,

$$q_{ij} = \frac{(1 + ||y_i - y_j||^2)^{-1}}{\sum_{k \neq l}(1 + ||y_k - y_l||^2)^{-1}}.$$

The primary downsides of t-SNE are its inherent randomness, unintuitive results, and sensitivity to hyperparameter calibration. The minimization of KL divergence is done using gradient descent methods with incorporated randomness to avoid stagnating at local minima. As a result, the output differs between runs of the algorithm. Hence, the traditional t-SNE workflow

 

often includes running the algorithm multiple times at various perplexities before choosing the best representation. t-SNE is also known to produce results that are not faithful to the true structure of the data, even when calibrated correctly. For example, cluster sizes and inter-cluster distances aren't always consistent with the original data [12]. Such artifacts of the t-SNE algorithm can be confused for significant structures by inexperienced users.

### UMAP

Uniform Manifold Approximation and Projection (UMAP, [5]) is another nonlinear dimension reduction method that has been rising in popularity. Originally introduced as a more computationally efficient alternative to t-SNE, UMAP is a powerful tool for visualizing high-dimensional data that requires user calibration. While its underlying ideology is completely different from that of t-SNE, the UMAP algorithm is very similar architecturally to the t-SNE algorithm—high-dimensional similarities are computed and the resulting representation is the set of low-dimensional points whose low-dimensional similarities best match the high-dimensional similarities. See [5] for details. The largest difference is UMAP's default initialization process. UMAP uses Laplacian eigenmaps to initialize the low-dimensional representation, which is then adjusted to minimize the cost function. Most t-SNE implementations use PCA during the initialization process. The initialization process is the primary benefit of the default implementation of UMAP, but t-SNE and UMAP have been shown to perform similarly with identical initializations [7]. Modern implementations of both algorithms are also comparable in speed.

UMAP shares similar disadvantages with t-SNE. It can create unfaithful representations that require experience to interpret and is sensitive to hyperparameter calibration [13].

## Methods

### Dimension reduction framework

Prior works quantitatively measure how well low-dimensional representations match the high-dimensional data. However, if we consider data as a composition of signal and noise, we must not reward capturing the noise. Therefore, we should be comparing the low-dimensional representation against the signal underlying our data, rather than the entirety of the data.

Suppose the underlying signal of our data is described by an $r$-dimensional matrix $Y \in \mathbb{R}^{n \times r}$. In the context of dimension reduction, the signal is often lower dimension than the original data. Let $p \geq r$ be the dimension of the original data set, and let $\text{Emb} : \mathbb{R}^r \to \mathbb{R}^p$ be the function that embeds the signal in data space. Define $Z = \text{Emb}(Y)$ to be the signal embedded in data space. We then assume the presence of random error. The original data can then be modeled by $Z + \epsilon$ for $\epsilon \sim N_p(0, \Sigma)$. The dimension reduction method $\varphi$ is applied to $Z + \epsilon$ to get a low-dimensional representation $X \in \mathbb{R}^{n \times q}$. See Fig 1.

### Reconstruction error functions

The remaining piece is a procedure for measuring dimension reduction performance. Suppose we have a reconstruction error function $f(D_1, D_2)$ that quantifies how well the data set $D_2$ represents the data set $D_1$. Prior works ([9, 10, 14–18]) use various reconstruction error functions to quantify performance; only, they study $f(Z + \epsilon, X)$ to measure how well the constructed representation $X$ represents the original data $Z + \epsilon$. We argue it is more appropriate to compare $X$ against the signal $Y$ by examining $f(Y, X)$.

Prior works in dimension reduction have suggested various quantitative metrics for measuring dimension reduction performance. In line with recent discussions of perplexity

 

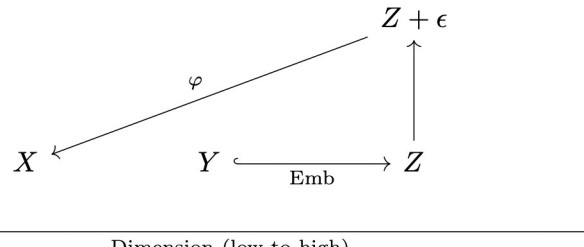

**Fig 1. Dimension reduction framework.** $Z + \epsilon$ represents the data constructed from an (often lower-dimensional) signal $Y$ embedded in data space. $X$ is the lower-dimensional embedding of $Z + \epsilon$ outputted by a DR technique. $X$ should be constructed to preserve $Y$, rather than $Z + \epsilon$.

([10, 14]), we employ two different metrics—one that measures local performance and one that measures global performance.

For local performance, we use a nearest-neighbor type metric called trustworthiness [19]. Let $n$ be the sample size and $r(i, j)$ the rank of point $j$ among the $k$ nearest neighbors of point $i$ in high dimension. Let $U_k(i)$ denote the set of points among the $k$ nearest neighbors of point $i$ in low dimension, but not in high dimension. Then

$$f_{trust}(D_1, D_2) = 1 - \frac{2}{nk(2n - 3k - 1)} \sum_{i=1}^{n} \sum_{j \in U_k(i)} [r(i, j) - k].$$

For each point, we are measuring the degree of intrusion into its $k$-neighborhood during the dimension reduction process. The quantity is then re-scaled, so that trustworthiness falls between 0 and 1 with higher values favorable. Trustworthiness is preferable to simply measuring the proportion of neighbors preserved because it's more robust to the choice of $k$. For very large values of $n$, we can get an estimate by only checking a random subsample of points $i_1, \ldots, i_m$. In this case,

$$f_{trust}(D_1, D_2) \approx 1 - \frac{2}{mk(2n - 3k - 1)} \sum_{l=1}^{m} \sum_{j \in U_k(i_l)} [r(i_l, j) - k].$$

Local performance is the primary concern when applying t-SNE and UMAP, so our experiments focus on maximizing trustworthiness.

For global performance, we use Shepard goodness [15]. Shepard goodness is the Spearman correlation, a rank-based correlation, between high and low-dimensional inter-point distances,

$$f_{Shep}(D_1, D_2) = \sigma_{Spearman}(||z_i - z_j||, ||\varphi(z_i) - \varphi(z_j)||).$$

Again for very large values of $n$, we can get an approximation by calculating the correlation between inter-point distances of a random subsample.

## Using this framework

When using this framework to model examples, three components must be specified: $Z + \epsilon$, $Y$, and Emb(). These elements describe the original data, the underlying signal, and the embedding of the signal in data space, respectively. When simulating examples, it's natural to start with the underlying signal $Y$ then construct $Z + \epsilon$ by attaching extra dimensions and adding

Gaussian noise. The Emb() function is then given by Emb($y$) = ($y$, 0, . . ., 0) so that

$$Z + \epsilon = [\,Y \quad | \quad 0\,] + \epsilon.$$

Practical examples are more tricky because we do not have the luxury of first defining $Y$. Instead, we are given the data $Z + \epsilon$ from which we must extract $Y$, or at least our best estimate. This process is dependent on the specifics of the problem and should be based on a priori knowledge of the data. If there is no specific signal of interest, a more general approach can be taken. We used a PCA projection of the data to represent the signal, $Y = \text{PCA}_r(Z + \epsilon)$, where $r$ is the dimension of the projection. For a reasonably chosen $r$, we expect the first $r$ principal components to contain most of the signal, while excluding most of the noise. Another advantage to using PCA is it gives rise to a natural Emb() function—the PCA inverse transform. If $Y$ is centered, then we may define

$$Z = \text{invPCA}_r(Y) = (Z + \epsilon)V_r V_r^T,$$

where $V_r \in \mathbb{R}^{p \times r}$ contains the first $r$ eigenvectors of $(Z + \epsilon)^T (Z + \epsilon)$ as column vectors.

## Results

### Simulated examples

We first looked at simulated examples with explicitly defined signal structures—three low-dimensional examples (Fig 2) and one high-dimensional example. The links example and the high-dimensional example are explored here. See Table 1 and S1 Appendix for the other simulated examples.

**Links data set.**   For the links example, the signal $Y$ consisted of two interlocked circles, each containing 250 points, embedded in three dimensions. $Z + \epsilon$ was constructed by adding seven superfluous dimensions and isotropic Gaussian noise. Various degrees of noise were tested ($sd$ = 0.5, 1, 1.5, 2, 2.5, 3).

t-SNE was run using the *R* package *Rtsne* [20] at varying perplexities. For each perplexity, the algorithm was run 40 times to mimic the ordinary t-SNE workflow. If the distinction between signal and noise was disregarded, a plot of $f_{\text{trust}}(Z + \epsilon, X)$ vs. perplexity could be used to maximize local performance. To avoid overfitting the noise, a plot of $f_{\text{trust}}(Y, X)$ vs. perplexity should be used instead. See Fig 3 for examples of these plots for the $sd$ = 1 case. Both plots depict an increase in local performance followed by a decrease as perplexity increases. This cutoff point, however, varies between the two plots. When comparing against the original data, the trustworthiness-maximizing representation was constructed with a perplexity of 40, which

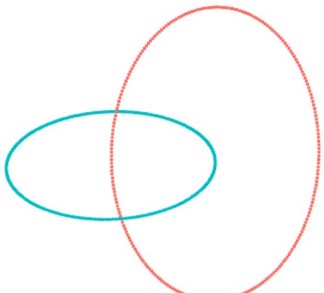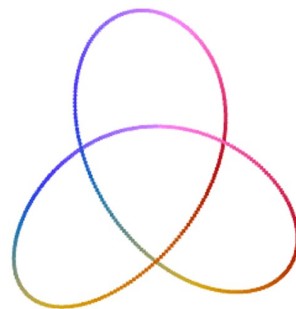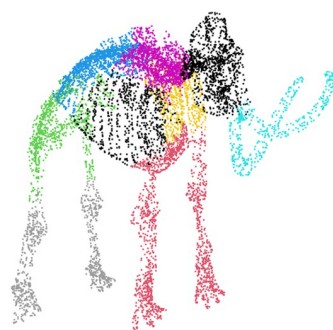

**Fig 2. Low-dimensional simulated examples.** Visualizations of the signals used to simulate data.

**Table 1. Summary of results.** Dimensionality details and optimal perplexity for each data set.

| Data Set | $n$ | $p$ | $r$ | Optimal Perplexity | |
|---|---|---|---|---|---|
| | | | | signal + noise | signal |
| Links [12] | 500 | 10 | 3 | 40 | 80 |
| Trefoil [12] | 500 | 10 | 3 | 35 | 100 |
| Mammoth [24] | 500 | 10 | 3 | 30 | 80 |
| High-Dimensional Clusters | 210 | 60 | 10 | 55 | 60 |
| scRNA-seq [21] | 864 | 500 | 5 | 40 | 120 |
| scRNA-seq [21] | 864 | 500 | 10 | 50 | 60 |
| CyTOF [22] | 5,000 | 30 | 5 | 50 | 110 |
| CyTOF [22] | 5,000 | 30 | 8 | 45 | 65 |
| Microbiome [23] | 280 | 66 | 5 | 50 | 90 |
| Microbiome [23] | 280 | 66 | 8 | 60 | 85 |

is consistent with the original authors' suggestion of 5 to 50 for perplexity [4]. When comparing against the signal, the trustworthiness-maximizing representation was constructed with a perplexity of 80.

With the signal structure known, we are also able to visually assess the trustworthiness-maximizing representations. Fig 4 shows the trustworthiness-maximizing representations for the $sd = 1$ case. Notice the larger perplexity was able to successfully separate the circles in the presence of noise, while the smaller perplexity was not. By using the signal as the frame of reference, our framework correctly rewarded the representation that was able to successfully separate the two links.

The same pattern held true for other levels of noise. The optimal perplexity was consistently larger when comparing against the signal, rather than the original data (Fig 5).

These results suggest larger perplexities perform better in the presence of noise, both quantitatively and qualitatively. We hypothesize t-SNE tends to overfit the noise when the perplexity is too small. Intuitively, small perplexities are more affected by slight perturbations of the data when only considering small neighborhoods around each point, leading to unstable representations. Conversely, larger perplexities lead to more stable representations that are more robust to noise.

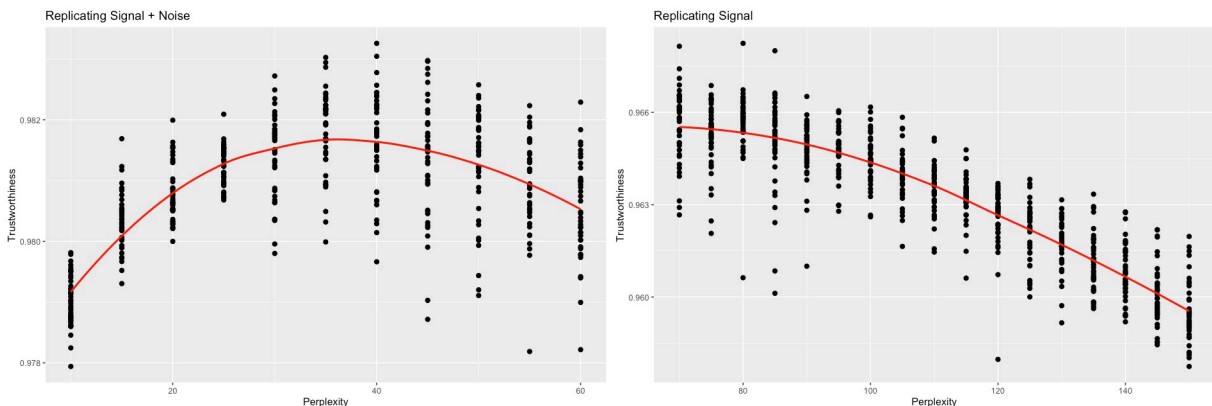

**Fig 3. Trustworthiness vs. perplexity (links $sd = 1$).** t-SNE outputs were calculated with varying perplexities. Local performance was measured via trustworthiness. The trustworthiness-maximizing perplexity was 40 when comparing against the original data, while the trustworthiness-maximizing perplexity was 80 when comparing against just the signal.

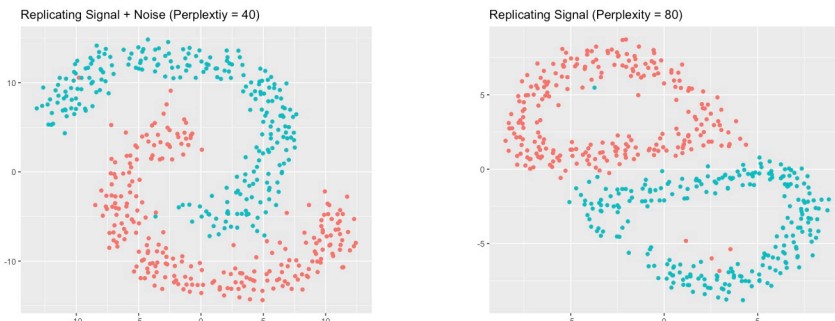

**Fig 4. Trustworthiness-maximizing representations (links *sd* = 1).** Trustworthiness-maximizing t-SNE outputs. Comparing against the signal resulted in a representation that better captured the two links.

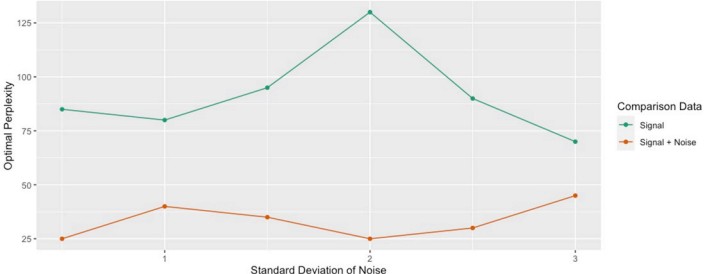

**Fig 5. Optimal perplexity (links).** The experiment was repeated at various levels of noise. For each level of noise, the trustworthiness-maximizing perplexity was recorded when comparing against the original data and the signal. The optimal perplexity was consistently greater when comparing against the signal.

**High-Dimensional Clusters.** The signal $Y$ consisted of seven Gaussian clusters, each containing 50 points, in seven dimensions. The clusters were drawn from multivariate normal distributions with mean $10e_i$ and random diagonal covariance matrices, where $e_i$ is the $i^{\text{th}}$ standard basis vector. The data set $Z + \epsilon$ was constructed from $Y$ by adding 53 superfluous dimensions and isotropic Gaussian noise to all 60 dimensions. Various degrees of noise were tested (*sd* = 2, 2.5, 3, 3.5, 4, 4.5).

When *sd* = 3, local performance peaked at different perplexities when changing the frame of reference (Fig 6). When comparing against the original data, trustworthiness was maximized at a perplexity of 55. When comparing against the signal, trustworthiness was maximized at a perplexity of 60. See Fig 7 for the trustworthiness-maximizing representations. Visually, both representations maintain the original clustering to some extent, but the higher-perplexity representation shows less mixing between the clusters and had a larger average silhouette width (0.178) than the lower-perplexity representation (0.121). This suggests the higher-perplexity representation better maintained the original clustering.

Fig 8 shows the optimal perplexities for different levels of noise. Again, the trustworthiness-maximizing perplexity was larger when comparing against the signal for all levels of noise.

## Practical examples

In addition to simulated data sets, we looked at three practical data sets: a single-cell RNA sequencing data set [21], a cytometry by time-of-flight (CyTOF) data set [22], and a

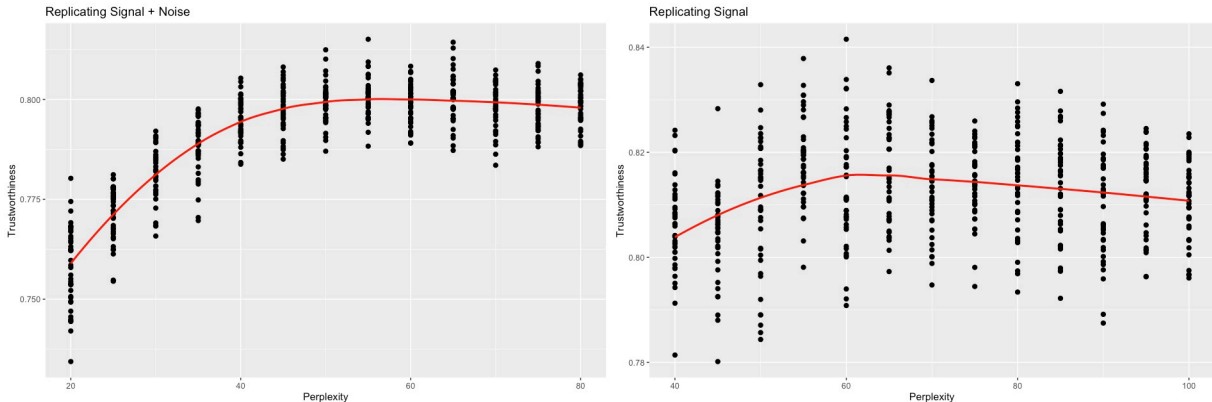

**Fig 6. Trustworthiness vs. perplexity (high-dimensional clusters *sd* = 3).** t-SNE outputs were calculated with varying perplexities. Local performance was measured via trustworthiness. The trustworthiness-maximizing perplexity was 55 when comparing against the original data, while the trustworthiness-maximizing perplexity was 60 when comparing against just the signal.

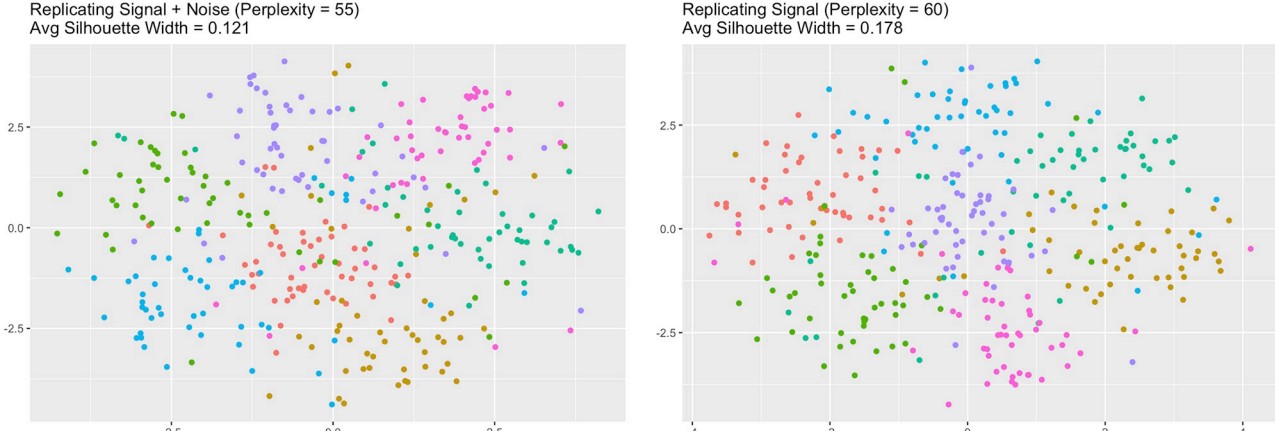

**Fig 7. Trustworthiness-maximizing representations (high-dimensional clusters *sd* = 3).** Trustworthiness-maximizing t-SNE outputs. Both outputs depict a similar clustering.

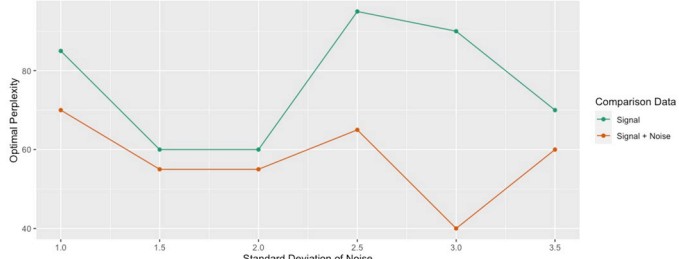

**Fig 8. Optimal perplexity (high-dimensional clusters).** The experiment was repeated at various levels of noise. For each level of noise, the trustworthiness-maximizing perplexity was recorded when comparing against the original data and the signal. The optimal perplexity was consistently greater when comparing against the signal.

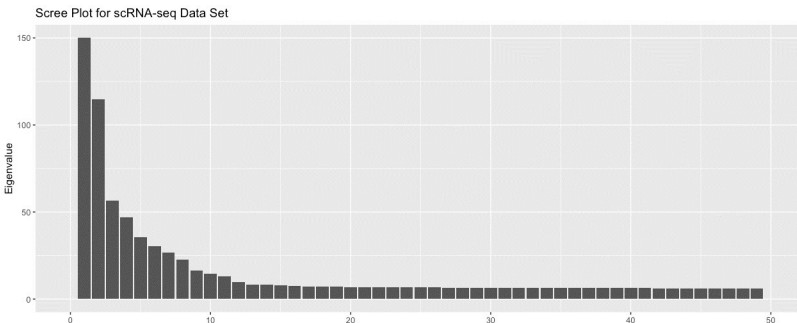

**Fig 9. Scree plot for scRNA-seq data set.** A PCA projection was used to extract the signal. To determine the appropriate number of dimensions for the projection, a scree plot was drawn.

microbiome data set [23]. For each data set, we compared the optimal perplexity for locally replicating the original data versus the estimated signal. We explore the scRNA-seq data set in detail here. The results of the other two practical examples can be found in Table 1. The details can be found in S1 Appendix.

The scRNA-seq data set was generated from induced pluripotent stem cells collected from three different individuals. The original data includes 864 units and 19,027 readings per unit. To process this zero-inflated count data, columns containing a large proportion of 0's (20% or more) were removed before a log transformation was applied. This step reduced the number of dimensions to 5,431. A PCA pre-processing step further reduced the number of dimensions to 500, which still retained 88% of the variance of the log-transformed data. Hence, the processed data set consisted of 864 observations in 500 dimensions, $Z + \epsilon \in \mathbb{R}^{864 \times 500}$. To determine the dimensionality of the signal, we drew a scree plot (Fig 9). Note, the first eigenvalue (2359.357) was cut to fit the plot. A conservative estimate is five dimensions, so $Y$ was extracted by taking the first five principal components, $Y = \text{PCA}_5(Z + \epsilon)$. We computed the t-SNE representations for perplexities ranging from 10 to 280. For each perplexity, 20 different t-SNE representations were computed.

As with the simulated examples, there is a difference in trend when switching the frame of reference (Fig 10). When compared against the original data, trustworthiness is maximized at

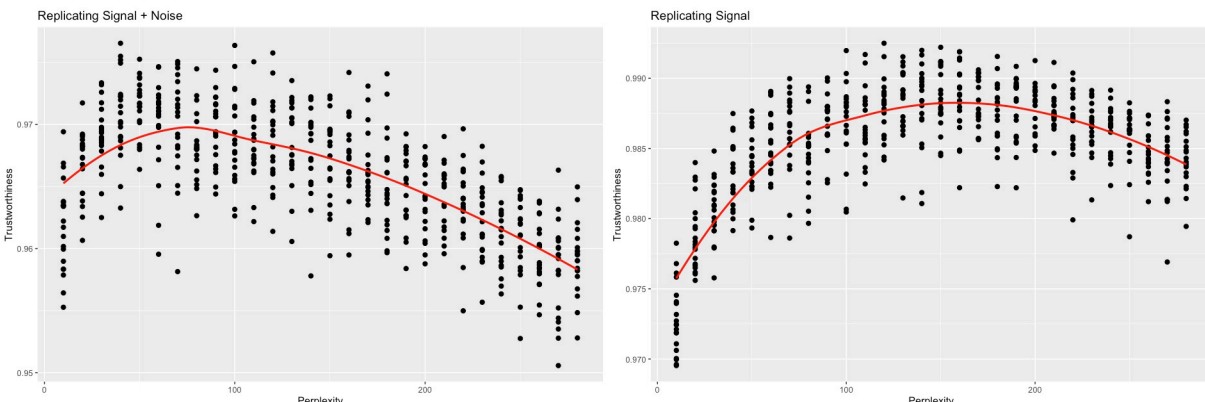

**Fig 10. Trustworthiness vs. perplexity for r = 5 (scRNA-seq).** t-SNE outputs were calculated with varying perplexities. Local performance was measured via trustworthiness. The trustworthiness-maximizing perplexity was 40 when comparing against the original data, while the trustworthiness-maximizing perplexity was 120 when comparing against just the signal.

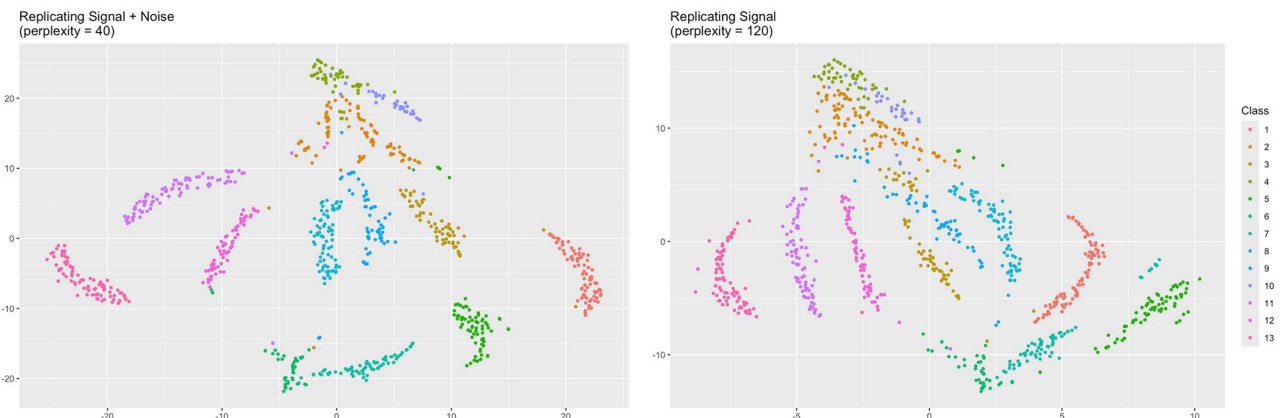

**Fig 11. Trustworthiness-maximizing representations for r = 5 (scRNA-seq).** Trustworthiness-maximizing t-SNE outputs. Both outputs depict a similar clustering with slightly varying cluster positioning. The perplexity = 40 representation depicts tighter clustering, but is outperformed in metrics measuring both local and global performance, suggesting the over-clustering and cluster positioning are misleading.

a perplexity of 40, which is consistent with [4]'s recommendation of 5 to 50. When compared against the signal, trustworthiness is maximized at a larger perplexity of 120, reinforcing the hypothesis that lower values of perplexity may be overfitting the noise.

Visual inspection of the trustworthiness-maximizing representations reveals the effect of increasing the perplexity (Fig 11). A hierarchical clustering of the high-dimensional data was computed, then projected onto the trustworthiness-maximizing representations. Both representations depict a similar structure, but the relative positioning of clusters differs. For example, the Class 1 cluster is the rightmost cluster in the perplexity = 40 representation, while the Class 5 cluster is the rightmost cluster in the perplexity = 120 representation. Furthermore, the left-to-right order of the Class 3, Class 8, and Class 9 clusters is reversed in both representations. Although relative positioning of clusters in t-SNE representations is often considered arbitrary, especially for low perplexities, the perplexity = 120 representation exhibits superior global performance. The perplexity = 40 representation has a Shepard goodness of 0.521 while the perplexity = 120 representation has a Shepard goodness of 0.788, suggesting the cluster positioning of the perplexity = 120 representation is more accurate than the cluster positioning of the perplexity = 40 representation.

In terms of local structure, the Class 3 and Class 7 clusters are better preserved in the perplexity = 40 representation, while the Class 12 cluster is better preserved in the perplexity = 120 representation. The perplexity = 40 representation also suggests the Class 2 cluster could potentially contain two separate clusters, but this is not consistent with the high-dimensional data according to the dendrogram and higher-order clusterings. The perplexity = 120 representation does not mislead in this way. Overall, the lower perplexity leads to tighter-knit clusters as expected. However, further investigation reveals the over-clustering may be unfaithful to the original data.

If we, instead, decide to be more conservative and use the first 10 principal components to represent the signal, we still see a similar trend (Figs 12 and 13). Trustworthiness still increases then decreases with perplexity. When compared against the original data, trustworthiness is maximized at a perplexity of 50 (Note the optimal perplexity when compared against the original data differed between the two experiments, even though it should theoretically be independent of the chosen signal dimension. This is due to the inherent randomness of the t-SNE algorithm). When compared against the signal, trustworthiness is maximized at a perplexity of

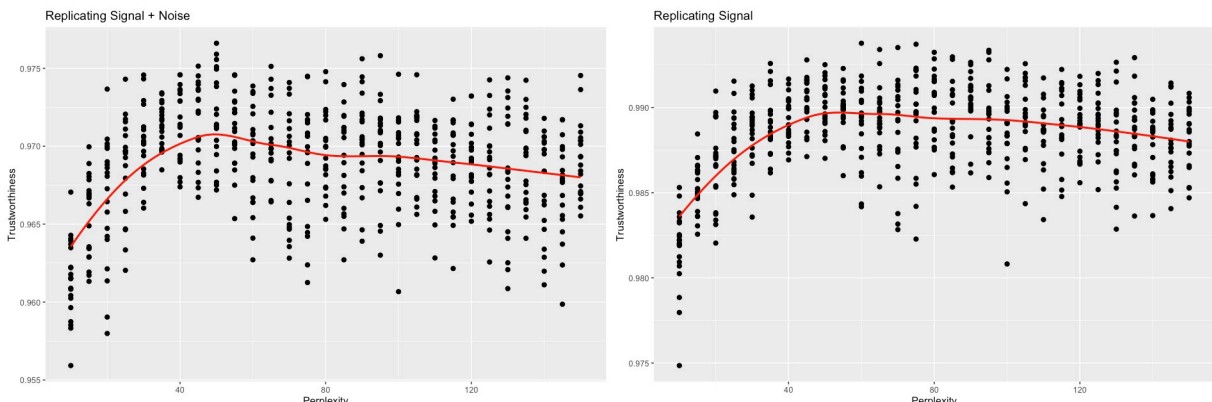

**Fig 12. Trustworthiness vs. perplexity for r = 10 (scRNA-seq).** t-SNE outputs were calculated with varying perplexities. Local performance was measured via trustworthiness. The trustworthiness-maximizing perplexity was 50 when comparing against the original data, while the trustworthiness-maximizing perplexity was 60 when comparing against just the signal.

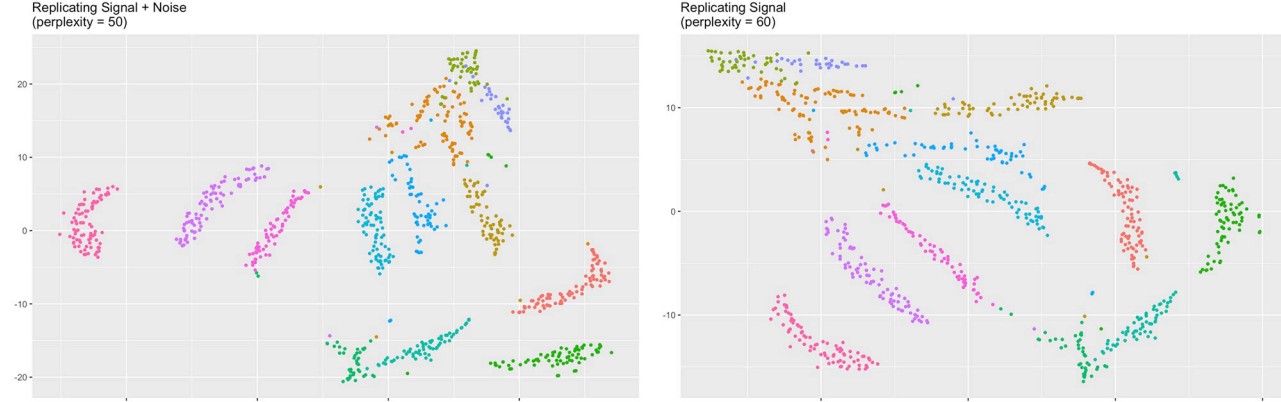

**Fig 13. Trustworthiness-maximizing representations for r = 10 (scRNA-seq).** Trustworthiness-maximizing t-SNE outputs. Both outputs depict a similar clustering with slightly varying cluster positioning.

60. By including five extra principal components in the signal, we're assuming the data contains less noise, allowing the model to be more aggressive during the fitting process.

## Summary of results

See Table 1 for a summary of the results. *n*, *p*, and *r* represent the sample size, dimension of the (post PCA-processed) data, and dimension of the extracted signal, respectively. The optimal perplexity when comparing against the signal was greater than the optimal perplexity when comparing against the original data for every example.

## UMAP and n_neighbors

If n_neighbors functions similarly to perplexity, we'd expect small values of n_neighbors to overfit the data as well. An identical experiment was run using the Python package *umap-learn* [5] on the scRNA-seq data. n_neighbor values ranging from 10 to 300 were tested and an n_neighbors value of 190 maximized trustworthiness when comparing against the original

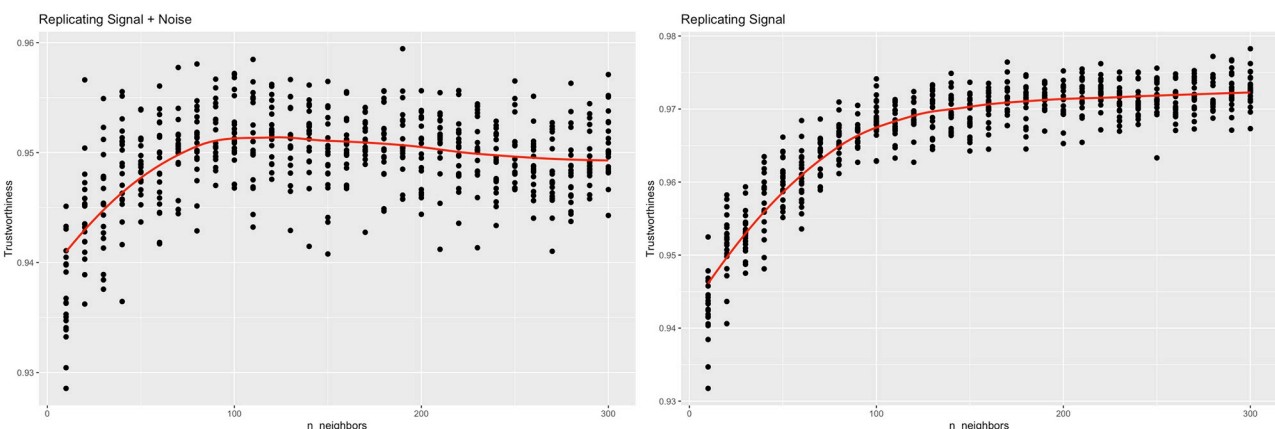

**Fig 14. Trustworthiness vs. n_neighbors for UMAP (scRNA-seq).** UMAP outputs were calculated with varying n_neighbors values. Local performance was measured via trustworthiness. The trustworthiness-maximizing n_neighbors was 190 when comparing against the original data, while the trustworthiness-maximizing n_neighbors was 300 when comparing against just the signal.

data, but an n_neighbors value of 300 maximized trustworthiness when comparing against the signal (Fig 14).

## Application

To apply this framework in practice, one must decide how to extract the signal from the data. The signal should include the features of the data one desires to retain throughout the dimension reduction process. When using a PCA projection to serve as the signal, one could draw a scree plot or employ a component selection algorithm such as parallel analysis [25] to determine the dimension of the signal.

With a signal constructed, it remains to compute t-SNE/UMAP outputs at varying perplexities/n_neighbors. It's recommended that at least a couple outputs are computed for each perplexity/n_neighbors to account for randomness in the algorithms. For each output, one must calculate the trustworthiness and Shepard goodness with respect to the signal. From there, one can choose the representation with the desired balance of local and global performance. A summary is given in Algorithm 1. Sample code is available at https://github.com/JustinMLin/DR-Framework/.

**Algorithm 1** Measuring Performance in the Presence of Noise

```
Require: original data Z + ϵ, perplexities {p₁, ..., pₘ} to test, and
neighborhood size k
1:  Y ⇐ PCAᵣ(Z + ϵ)
2:  perplexities ⇐ {p₁, ..., pₘ}
3:  for perplexity in perplexities do
4:    loop
5:      X_tsne ⇐ Rtsne(Z + ϵ, perplexity)
6:      trust ⇐ trustworthiness(Y, X_tsne, k)
7:      shep ⇐ Shepard_goodness(Y, X_tsne)
8:    end loop
9:  end for
10: Plot trustworthiness and Shepard goodness values
11: Choose output with desired balance of local and global performance
```

It is worth noting that computational barriers may arise, especially for very large data sets. To alleviate such issues, trustworthiness and Shepard goodness can be approximated by

subsampling before calculation. Furthermore, t-SNE and UMAP are generally robust to small changes in perplexity and n_neighbors, so checking a handful of values is sufficient. If computing multiple low-dimensional representations is the limiting factor, one can try calibrating the hyperparameters for a subsample before extending to the full data set. [26] found that embedding a $\rho$-sample, where $\rho \in (0, 1]$ is the sampling rate, with perplexity Perp′ gives a visual impression of embedding the original data with perplexity Perp $= \frac{\text{Perp}'}{\rho}$. With these concessions, applying this framework to calibrate hyperparameters should be feasible for data sets of any size.

## Case study

To demonstrate how one might apply this framework, we walk through a detailed case study on a modern scRNA-seq data set.

### Data

Cryopreserved human peripheral blood mononuclear cells (PBMCs) from a healthy female donor aged 25 were obtained by 10x Genomics from AllCells. Granulocytes were removed by cell sorting, followed by nuclei isolation. Paired ATAC and Gene Expression libraries were generated from the isolated nuclei and sequenced. See [27] for details.

### Pre-processing

Pre-processing was completed using the *R* package *BPCells* and the steps followed the provided tutorial [28] closely. Low quality cells (those that did not meet the required number of RNA reads, the required number of ATAC reads, or TSS Enrichment cutoffs) were filtered out before a matrix normalization was applied. The cleaned dataset contained 2,600 cells and 1,000 genes. The number of dimensions was then reduced to 500 using PCA, which retained 86% of the original variance. The processed data set to be analyzed contained 2,600 observations in 500 dimensions, $\mathbb{Z} + \epsilon \in \mathbb{R}^{2,600 \times 500}$.

### Determining the signal

To determine the number of signal dimensions, a scree plot was drawn (Fig 15). The first eigenvalue was approximately 188 but was trimmed to fit the plot. Four dimensions, a relatively conservative estimate, were chosen to represent the signal, $Y \in \mathbb{R}^{2,600 \times 4}$.

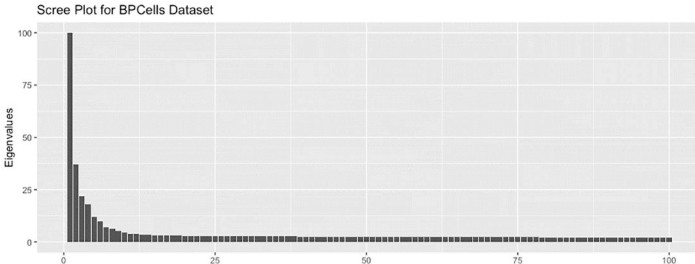

**Fig 15. Scree plot for PBMC data set.** A PCA projection was used to extract the signal. To determine the appropriate number of dimensions for the projection, a scree plot was drawn.

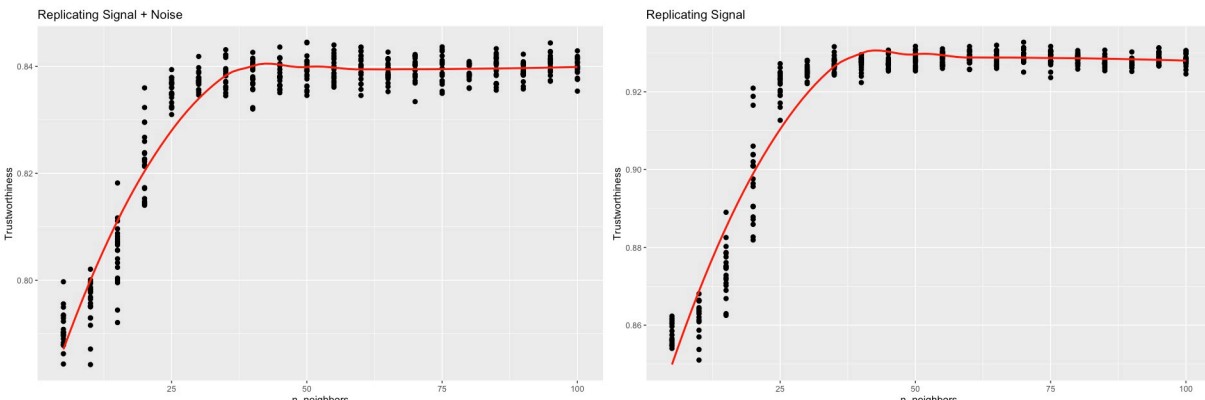

**Fig 16. Trustworthiness vs. n_neighbors for UMAP (PBMC).** UMAP outputs were calculated with varying n_neighbors values. Local performance was measured via trustworthiness. The trustworthiness-maximizing n_neighbors was 50 when comparing against the original data, while the trustworthiness-maximizing n_neighbors was 70 when comparing against just the signal.

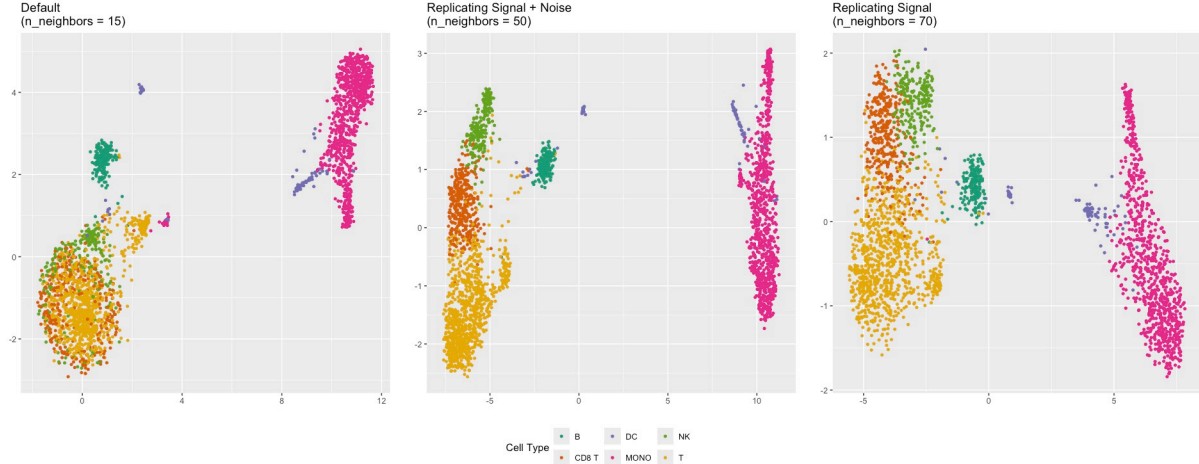

**Fig 17. Cell types (PBMC).** UMAP representations for different values of n_neighbors. The cell types were assigned through study of known marker genes. The n_neighbors = 50 and n_neighbors = 70 representations did the best job separating the different cell types. The n_neighbors = 50 representation is more tightly clustered than the n_neighbors = 70 representation. The relative positioning of the NK and CD8 T cells differs between the n_neighbors = 50 and n_neighbors = 70 representations.

## Results

UMAP was applied with multiple values of n_neighbors. 20 representations were computed for each value, and trustworthiness was measured with respect to both the entire data and the signal. Trustworthiness was maximized at a n_neighbors value of 50 when comparing against the entire data and a value of 70 when comparing against the signal (Fig 16). Cell types (B, T, Monocyte, NK, Dendritic cell, CD8 T) were assigned to each cluster by exploring marker genes (Fig 17). See [28] for details.

## Analysis

In all three representations, the primary division of cell types is between monocytes and some of the dendritic cells vs. the T, B, CD8 T, and NK cells. In the default UMAP representation,

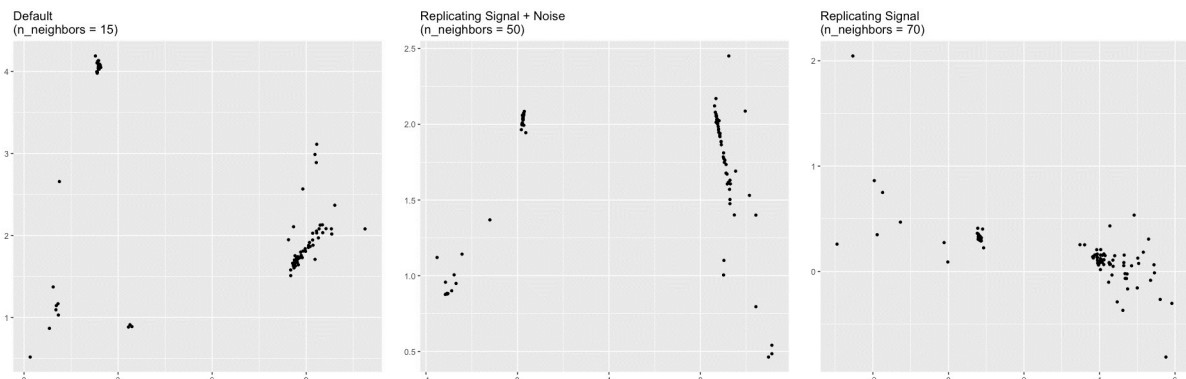

**Fig 18. Plot of dendritic cells (PBMC).** Dendritic cells (DC) extracted from the UMAP representations constructed with different values of n_neighbors. The n_neighbors = 15 and n_neighbors = 50 representations show two clusters, while the n_neighbors = 70 representation may be showing three clusters.

the B cells form a cluster that is quite distinct from the T, CD8 T, and NK cells. As we increase n_neighbors to 50 and 70, the B cell cluster moves closer to the T/CD8 T/NK cell cluster. The closer proximity of the B cells to the T, CD8 T, and NK cells in the n_neighbors = 70 representation is consistent with the over-arching categorization of T, CD8 T, NK, and B cells as lymphocytes, as opposed to monocytes.

Perhaps a starker difference between the representations concerns the dendritic cells (DCs). In the n_neighbors = 15 and n_neighbors = 50 representations, there are three distinct clusters of DCs, whereas there are only two in the n_neighbors = 70 representation (Fig 18). Principal component analysis of the DCs alone suggests that the DCs are either two clusters, one of which is more diffuse than the other, or three clusters, two of which are fairly close together (Fig 19). Standard metrics for determining the number of clusters suggest the same. The silhouette width metric suggests two clusters (Fig O in S1 Appendix), while the gap statistic suggests three (Fig P in S1 Appendix). However, the three-cluster solution given by k-means and visual inspection of the principal components plot does not align with the three clusters in the n_neighbors = 15 or n_neighbors = 50 representation. The green and orange clusters are

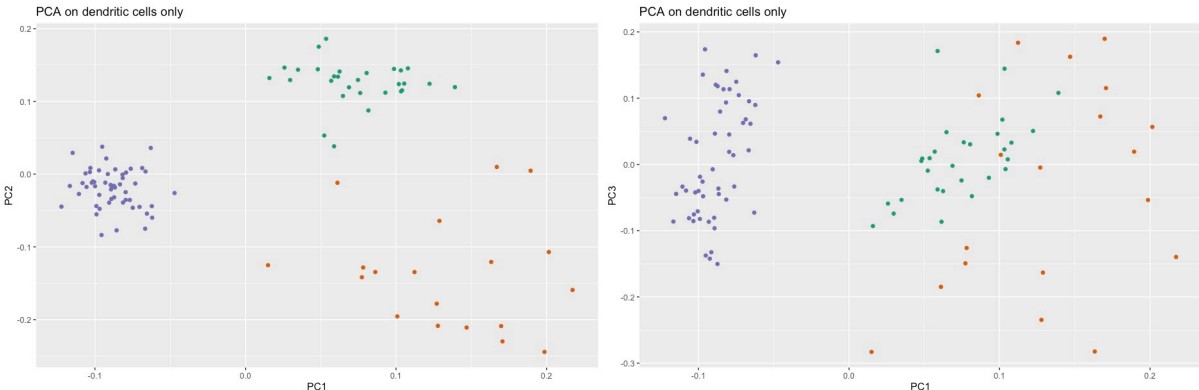

**Fig 19. PCA applied to dendritic cells (PBMC).** PCA was applied to the subset of dendritic cells. The first two principal components seem to imply the dendritic cells belong to three different clusters. The points were assigned according to a three-cluster k-means clustering upon the PCA projection.

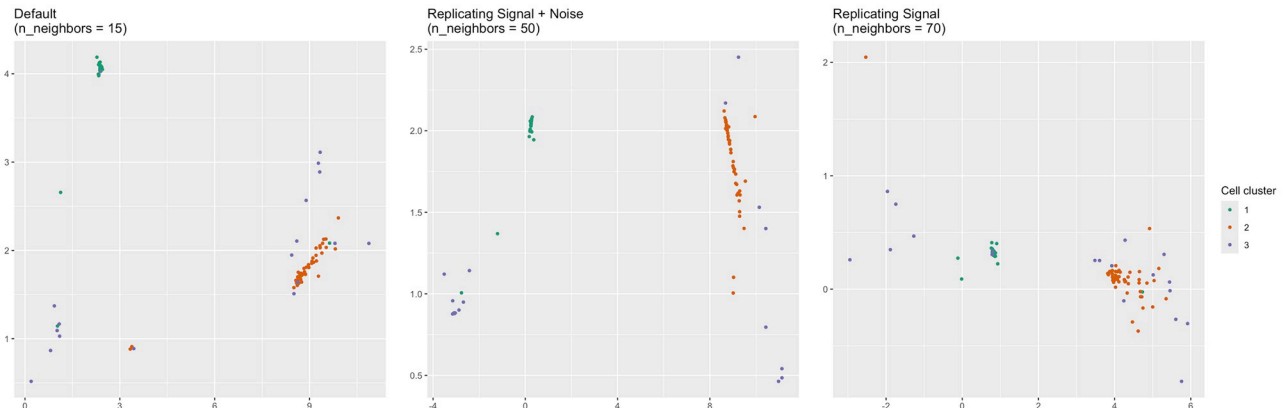

**Fig 20. Dendritic cells colored according to PCA projection (PBMC).** Dendritic cells (DC) extracted from the UMAP representations colored according to the k-means clustering upon the PCA projection of the dendritic cells. The n_neighbors = 70 representation separated the purple points the least among the three representations.

represented faithfully, but the third, more diffuse, purple cluster is split across two DC clusters in the n_neighbors = 15 and n_neighbors = 50 representations (Fig 20). The degree of separation is lesser in the n_neighbors = 70 representation. Therefore, the n_neighbors = 15 and n_neighbors = 50 representations inaccurately represent the dendritic cells in a way that the n_neighbors = 70 representation does not.

## Discussion

We have illustrated the importance of acknowledging noise when performing dimension reduction by studying the roles perplexity and n_neighbors play in overfitting data. When using the original data to calibrate perplexity, our experiments agreed with perplexities previously recommended. When using the signal, however, our experiments indicated that larger perplexities perform better. Low perplexities/n_neighbors lead to overly-flexible models that are heavily impacted by the presence of noise, while higher perplexities/n_neighbors exhibit better performance due to increased stability. These considerations are especially important when working with heavily noised data, which are especially prevalent in the world of single-cell transcriptomics [29].

We have also presented a framework for modeling dimension reduction problems in the presence of noise. This framework can be used to study other hyperparameters and their relationships with noise. In the case when a specific signal structure is desired, this framework can be used to determine which dimension reduction method best preserves the desired structure. Further works should explore alternative methods for extracting the signal in a way that preserves the desired structure.

## Supporting information

**S1 Appendix. Supporting information file, including extra figures.**
(PDF)

## Author Contributions

**Conceptualization:** Justin Lin, Julia Fukuyama.

**Data curation:** Justin Lin.

**Formal analysis:** Justin Lin.

**Investigation:** Justin Lin, Julia Fukuyama.

**Methodology:** Justin Lin, Julia Fukuyama.

**Resources:** Julia Fukuyama.

**Software:** Justin Lin.

**Supervision:** Julia Fukuyama.

**Validation:** Julia Fukuyama.

**Visualization:** Justin Lin.

**Writing – original draft:** Justin Lin.

**Writing – review & editing:** Julia Fukuyama.

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
