## [Decision Letter · Decision Letter 0]

22 Mar 2024

Dear Mr. Lin,

Thank you very much for submitting your manuscript "Calibrating dimension reduction hyperparameters in the presence of noise" for consideration at PLOS Computational Biology.

As with all papers reviewed by the journal, your manuscript was reviewed by members of the editorial board and by several independent reviewers. In light of the reviews (below this email), we would like to invite the resubmission of a significantly-revised version that takes into account the reviewers' comments.

It is important that both reviewer 1 and 2's comments are thoroughly addressed including applications to a real dataset.

We cannot make any decision about publication until we have seen the revised manuscript and your response to the reviewers' comments. Your revised manuscript is also likely to be sent to reviewers for further evaluation.

Sincerely,

Sushmita Roy, Ph.D.

Section Editor

PLOS Computational Biology

Sushmita Roy

Section Editor

PLOS Computational Biology

Reviewer's Responses to Questions

**Comments to the Authors:**

Reviewer #1: The authors present analyses and some recommendations for tuning dimension reduction algorithm hyperparameters, with a focus on the effect of noise. They present examples on simulated and real data showing that higher perplexity (for t-SNE) is typically desired to faithfully reflect the underlying signal. Overall, the authors convincingly demonstrate the importance of considering the effect of noise in measurements and the effect of perplexity on faithfully capturing underlying signal. I have the following comments:

Major:

- For simulated examples, it would be useful to sweep over signal-to-noise ratio (by adjusting the gaussian noise) and plotting the optimal perplexity as the SNR changes. This can also be performed for practical examples by sweeping over PCs. E.g. I would recommend keeping one of Fig 6/7 and add a plot that shows the optimal perplexity as a function of PCs.

- At the moment, dimension reduction is a polarizing topic in the field of single-cell genomics, and primarily single-cell transcriptomics. This paper has the potential to add much needed nuance to this discussion. My recommendation would be to add a detailed case-study that applies the authors proposed recommendations (in Section 5) and walks the reader through a quantitative and qualitative evaluation of the results. For this, I would suggest working with one more more recent 10x Genomics based single-cell RNA-seq dataset (instead of the Fluidigm dataset from Tung et al) and use UMAP, as it's the most commonly used method for scRNA-seq data. The authors should apply their recommendations to these datasets, and show how changing the n_neighbors parameter changes trustworthiness and Shephard goodness, as well as provide a qualitative assessment of the differences between the UMAPs (which should be presented in the main text). Walking the reader through the key observations and decisions would empower the users of such methods to make reasonable choices in their work, and could be a very useful contribution.

Minor:

- In Perplexity vs Trustworthiness plots such as 3b, the color is redundant with the x-axis. My recommendation would be to remove the color, and instead add a Loess curve to highlight the trend.

- Please add references to specific supplementary plots, e.g. in “See Supporting Information for plots."

- Line 66, 215 typo: inherit -> inherent

Reviewer #2: This paper considers an aspect of dimensionality reduction for visualization that is often overlooked, at least in a direct sense, by many methods: the role of noise. At least in their original presentations, the popular methods of t-SNE and UMAP do not consider the role of noise, especially in the context of choosing hyperparameters. Since real data are often noisy, this is important to consider. However, the novelty and analysis in the paper are lacking in many important ways.

First, while t-SNE and UMAP do not consider the noise in the data, many other methods do. In particular, methods based on the diffusion maps framework denoise the data as part of the dimensionality reduction. These methods include, but are not limited to, diffusion maps [R1], PHATE [R2], MultiScale PHATE [R3], RF-PHATE [R4], EIG [R5], and DIG [R6]. The amount of diffusion is often considered in the context of denoising as well (see [R2]). So these references should be included and the authors' claims that noise is not considered in dimensionality reduction should be softened, if not eliminated entirely. This diminishes the claimed novelty of the proposed work.

Second, the authors' claims about the lack of consideration of noise isn't even entirely true about t-SNE and UMAP, especially in the biological world. PCA is often performed as a preprocessing step for both methods, which does end up denoising the data to some degree. In fact, the authors propose using PCA as an embedding method for determining the noise level. This combined with the previous point make it difficult to see what novelty is added here beyond common practice.

Based on the authors' results, they do give different guidelines for hyperparameters for t-SNE and UMAP than is commonly suggested. These are obtained by effectively optimizing the trustworthiness and/or Shepard goodness with respect to the PCA representation. However, the authors never show any actual t-SNE or UMAP visualizations that compare the different hyperparameters. Thus it is not clear if the new suggested values offer any improvement over the old ones from a visual perspective. 

Section 3.3: Modeling the embedding function using the PCA inverse transform seems very limiting and also seems to undermine the proposed approach as the optimal dimensionality reduction in this setting would be to do PCA. It's not clear why anyone would want to do t-SNE or UMAP to reduce dimensions if PCA is the optimal embedding function. It seems we would need more complex embedding functions to justify the use of t-SNE or UMAP (or anything besides PCA). Because of this potential disparity, it is not clear that using PCA to represent the "true" signal would correlate well with good hyperparameters for methods like t-SNE or UMAP. Hence the need for visualizations to verify this, especially in the case for the simulated and simple datasets.

Some other points:

Given this is PLOS Computational Biology, the introduction should talk more about biology uses for visualization than it currently does. In addition to the references above, the authors should reference [R7-R9, etc.].

Line 66: many other downsides of t-SNE exist that should be discussed. See [R10]. UMAP inherits many of these as well.

Section 2.2: It has been shown that the main benefit that UMAP has over t-SNE is its initialization using Laplacian eigenmaps. Initializing t-SNE with Laplacian eigenmaps gives similar results to UMAP [R9]. This is worth mentioning. Perhaps it is worth exploring if the recommendations change when t-SNE is initialized using Laplacian eigenmaps instead of the default random initialization.

A better alternative to the Shepard goodness is the Mantel test. The Mantel test takes into account the correlations between distances, which is largely ignored in the Shepard goodness.

Section 4.1: How are the # of PCA dimensions chosen for these experiments?

A little bit of proofreading is needed in the paper.

[R1] Coifman and Lafon, "Diffusion Maps", ACHA, 2006.

[R2] Moon et al., "Visualizing transitions and structure for biological data exploration," Nature Biotechnology, 2019.

[R3] Kuchroo et al., "Multiscale PHATE identifies multimodal signatures of COVID-19", Nature Biotechnology, 2022.

[R4] Rhodes et al., "Gaining biological insights through supervised data visualization," bioRxiv, 2024.

[R5] Talmon and Coifman, "Empirical intrinsic geometry for nonlinear modeling and time series filtering," PNAS, 2013.

[R6] Duque et al, "Visualizing high dimensional dynamical processes," MLSP, 2019.

[R7] Amir et al., "viSNE enables visualization of high dimensional single cell data and reveals phenotypic heterogeneity of leukemia," Nature Biotechnology, 2013.

[R8] Becht et al, "Dimensionality reduction for visualizing single-cell data using UMAP," Nature Biotechnology, 2019.

[R9] Kobak and Linderman, "Initialization is critical for preserving global data structure in both t-SNE and UMAP," Nature Biotechnology, 2021.

[R10] Wattenberg et al, "How to use t-SNE effectively," Distill, 2016.

**Have the authors made all data and (if applicable) computational code underlying the findings in their manuscript fully available?**

Reviewer #1: Yes

Reviewer #2: Yes

PLOS authors have the option to publish the peer review history of their article (what does this mean?). If published, this will include your full peer review and any attached files.

Reviewer #1: No

Reviewer #2: No
---

## [Decision Letter · Decision Letter 1]

2 Jul 2024

Dear Mr. Lin,

Thank you very much for submitting your manuscript "Calibrating dimension reduction hyperparameters in the presence of noise" for consideration at PLOS Computational Biology.

As with all papers reviewed by the journal, your manuscript was reviewed by members of the editorial board and by several independent reviewers. In light of the reviews (below this email), we would like to invite the resubmission of a significantly-revised version that takes into account the reviewers' comments.

Reviewer 2 still has some outstanding issues, so please address these comments fully.

We cannot make any decision about publication until we have seen the revised manuscript and your response to the reviewers' comments. Your revised manuscript is also likely to be sent to reviewers for further evaluation.

Sincerely,

Sushmita Roy, Ph.D.

Section Editor

PLOS Computational Biology

Sushmita Roy

Section Editor

PLOS Computational Biology

Reviewer's Responses to Questions

**Comments to the Authors:**

Reviewer #1: The authors have satisfactorily addressed all my comments.

Reviewer #2: The authors have largely addressed my concerns. The main thing I would still like to see is t-SNE visualizations for the datasets used in Section 4.2. The authors provided visualizations for the simulated data and the new dataset in Section 6, but not for the datasets in Section 4.2. Again, since the authors are claiming that the proposed approach improves visualization, I expect to see visual evidence of that for all of the datasets.

Minor points:

Technically, the Mantel test can be applied with the Spearman correlation and not just the Pearson correlation. I'll leave it up to the authors to decide if they wish to use it.

Figure 18 appears to be miscaptioned and could probably be combined with Figure 16.

**Have the authors made all data and (if applicable) computational code underlying the findings in their manuscript fully available?**

Reviewer #1: Yes

Reviewer #2: Yes

PLOS authors have the option to publish the peer review history of their article (what does this mean?). If published, this will include your full peer review and any attached files.

Reviewer #1: No

Reviewer #2: No
---

## [Decision Letter · Decision Letter 2]

4 Aug 2024

Dear Mr. Lin,

Thank you very much for submitting your manuscript "Calibrating dimension reduction hyperparameters in the presence of noise" for consideration at PLOS Computational Biology. As with all papers reviewed by the journal, your manuscript was reviewed by members of the editorial board and by several independent reviewers. The reviewers appreciated the attention to an important topic. Based on the reviews, we are likely to accept this manuscript for publication, providing that you modify the manuscript according to the review recommendations.

Please address the remaining outstanding comment from reviewer 2.

Sincerely,

Sushmita Roy, Ph.D.

Section Editor

PLOS Computational Biology

Sushmita Roy

Section Editor

PLOS Computational Biology

Reviewer's Responses to Questions

**Comments to the Authors:**

Reviewer #2: I have one final comment. On page 11, the discussion of Figure 11 discusses the relative position of clusters and how that changes with different values of perplexity. An important point is that the relative position of clusters in t-SNE embeddings is arbitrary and meaningless. Thus the authors should either eliminate this discussion or clarify this point.

**Have the authors made all data and (if applicable) computational code underlying the findings in their manuscript fully available?**

Reviewer #2: None

PLOS authors have the option to publish the peer review history of their article (what does this mean?). If published, this will include your full peer review and any attached files.

Reviewer #2: No

Figure Files:

Data Requirements:

Reproducibility:

References:

---

## [Decision Letter · Decision Letter 3]

19 Aug 2024

Dear Mr. Lin,

We are pleased to inform you that your manuscript 'Calibrating dimension reduction hyperparameters in the presence of noise' has been provisionally accepted for publication in PLOS Computational Biology.

Best regards,

Sushmita Roy, Ph.D.

Section Editor

PLOS Computational Biology

Sushmita Roy

Section Editor

PLOS Computational Biology

Reviewer's Responses to Questions

**Comments to the Authors:**

Reviewer #2: All concerns have been adequately addressed.

**Have the authors made all data and (if applicable) computational code underlying the findings in their manuscript fully available?**

Reviewer #2: None

PLOS authors have the option to publish the peer review history of their article (what does this mean?). If published, this will include your full peer review and any attached files.

Reviewer #2: No

---

## [Editor Report · Acceptance letter]

6 Sep 2024

PCOMPBIOL-D-24-00178R3 

Calibrating dimension reduction hyperparameters in the presence of noise

Dear Dr Lin,

I am pleased to inform you that your manuscript has been formally accepted for publication in PLOS Computational Biology. Your manuscript is now with our production department and you will be notified of the publication date in due course.

With kind regards,

Zsofia Freund
